# Analysis of Laser Sensors and Camera Vision in the Shoe Position Inspection System

**DOI:** 10.3390/s21227531

**Published:** 2021-11-12

**Authors:** Jaromír Klarák, Ivan Kuric, Ivan Zajačko, Vladimír Bulej, Vladimír Tlach, Jerzy Józwik

**Affiliations:** 1Faculty of Mechanical Engineering, University of Žilina, 010 26 Žilina, Slovakia; ivan.kuric@fstroj.uniza.sk (I.K.); ivan.zajacko@fstroj.uniza.sk (I.Z.); vladimir.bulej@fstroj.uniza.sk (V.B.); vladimir.tlach@fstroj.uniza.sk (V.T.); 2Institute of Informatics, Slovak Academy of Sciences, 845 07 Bratislava, Slovakia; 3Department of Production Engineering, Mechanical Engineering Faculty, Lublin University of Technology, 20-618 Lublin, Poland; j.jozwik@pollub.pl

**Keywords:** shoe inspection, polynomial regression, point cloud, laser sensor, shoe, camera vision

## Abstract

Inspection systems are currently an evolving field in the industry. The main goal is to provide a picture of the quality of intermediates and products in the production process. The most widespread sensory system is camera equipment. This article describes the implementation of camera devices for checking the location of the upper on the shoe last. The next part of the article deals with the analysis of the application of laser sensors in this task. The results point to the clear advantages of laser sensors in the inspection task of placing the uppers on the shoe’s last. The proposed method defined the resolution of laser scanners according to the type of scanned surface, where the resolution of point cloud ranged from 0.16 to 0.5 mm per point based on equations representing specific points approximated to polynomial regression in specific places, which are defined in this article. Next, two inspection systems were described, where one included further development in the field of automation and Industry 4.0 and with a high perspective of development into the future. The main aim of this work is to conduct analyses of sensory systems for inspection systems and their possibilities for further work mainly based on the resolution and quality of obtained data. For instance, dependency on scanning complex surfaces and the achieved resolution of scanned surfaces.

## 1. Introduction

Industry 4.0 is a new trend in the world being very popular due to new possibilities of solving specific issues occurring in modern factories mainly in the field of automation, inspection, data collections, artificial intelligence, etc. The progress number of articles about Industry 4.0 indexed in Scholar is described in [1]. The increase of papers in 2020 are associated with the keywords Artificial Intelligence, Internet of Things, Digital Twin, and Blockchain. The other survey set in four companies from the automotive area in Poland is described in [2], where it interpreted the opinion production of employees, engineers, and managers to Industry 4.0 and implementation in the factory. The feedback can be represented as positive, and these methods included in Industry 4.0 have a good background for future growth. An interesting study carried out among 65 Italian companies is described in [3], where the implementation methods of Industry 4.0 are studied. The study involved 39 companies, which have been classified as the medium digital level and 13 companies have been classified as the high digital level. Results in advantages from Industry 4.0 are mainly in quality in the way of reduction of errors, logistics, and time-saving, which could be represented as crucial economic benefits. Unfortunately, the results also showed a lack of awareness of environmental sustainability from companies as European green aims. The application of measurement systems as part of Industry 4.0 is described in [4]. There are express advantages and benefits in applying these methods mainly in the quality and economic fields. The authors in [5] are describing the system TACNET 4.0 based on 5G communication technology that provides wireless connection in complex systems including the sensors, actuators, robots, etc. 

Based on new technologies in wireless communication, sensing technologies, data collection, cloud, and evaluating data the development of Industry 4.0 is still in progress. These fields make the background for smart, and more efficient manufacturing processes which are reflected in production expenses and in improving productivity and managing. It is essential to develop the system of Internet of Things (IoT) for collection, processing, evaluating, storage, and usage of data from sensory systems integrated in a smart factory [6,7]. The other possibility to gain data from the manufacturing is by implementing commercial solutions based on 3D camera vision, such as Zivid Two or Zivid One+ from Zivid company. The application is for instance described in [8], where it demonstrates the work of the surgical Da Vinci robot with a 3D view on pegboard. Other similar commercial solutions are MotionCam-3D and PhoXi 3D Scanner GEN 2 from the Photoneo company. The example of the application the PhoXi 3D Scanner is described in [9], where it demonstrates the manipulation of objects from Picking Bin to Deposit Bin by robot ABB Yumi.

There are many benefits in such modern trend integrations. In the fashion industry or shoe industry, it is very perspective to collect data in order to evaluate the quality during manufacturing processes, mainly to declare and prove the guaranty of quality to customers by data from sensor systems employed in the factory [10]. To support the presented high quality of branded products by unquestionable data from the manufacturing process. Based on the data from sensors integrated into the inspection system, it is also possible to take an action within the manufacturing process quickly to prevent failures, to capture unsuitable activity, and to remove it from the manufacturing process.

Nowadays, end-consumer-oriented production is focused on design, ergonomics, quality, ecology, price, etc. Manufacturers in the shoe industry, such as one part of the fashion industry, aim to fulfil the expectations of end-consumers. The internal criteria of producers are also the price of intermediate products, amount of waste, time of production, etc. The terms of products criteria are also possible to be described in a different way [11,12], while the term of quality is definable in various ways. For this paper, one of the considerable measurable aspects of the quality is the uniformity of a pair of shoes. Every pair should be described as a mirror pair, where differences are negligible and for end-consumer is finally elusive. These possible differences originate mainly in manual operation during production due to the subjectivity of workers, especially in performing a sewing, placement, or other operations including the addition or removing of the materials during the manufacturing process. The differences also arise in the automated process. In this case, their reason is in the wrong input of the material, a machine malfunction or other malfunction aspect of production processes. Based on analysis of the manufacturing process, it is possible to define nodes, where inspection systems may be established to perform capturing characteristics of object and condition in the manufacturing process and analysis of these data and evaluation of production quality in a specific node. The outcome of this analysis is information about the process of a particular item of production, the possibility of repair of the item, or the removal of the item from the production process if the defect is non-repairable. The implementation of inspection systems can be focused on quality but also on the environmental aspect. Within the quality parameters, there should be considered monitoring of the attributes affecting the quality of the final product. In the ecology aspect, it is in manner of removing any non-repairable materials used in production and in the final product, to decrease the amount of waste arising from the process of manufacturing.

The production process consists of preparing the parts of the shoe, mainly textile or leather materials, for instance, the quarter or vamp, etc.; preparing these soft parts is mainly automated or machined. Then there comes the cutting of the outer parts and sewing them, forming the integral entirety creating an upper (Figure 1) or shoe without a sole or an insole. Sewing parts into the integral entirety is often performed manually by human workers. The quality of this process is, therefore, human-dependent. The shoe upper is in the next step placed over the shoe last. Being all prepared; the next operation is of technological character; it is the production of a sole. Very often it is performed by injection molding technology, applicating polyurethane materials to mold. It is possible to produce the soles separately or to applicate them directly on the upper placed on the last. In case of application of the sole to the shoe upper, the last with the upper is placed in a mold. This mold is filled by the polyurethane material or any other materials used for sole formation. After this technological process, a sole contains surplus material, which is necessary to be removed. Surplus materials arise from leaks occurring in the mold. The process of removing surplus material is mainly performed by a person. Automation of this process is demanding because of the non-exactly the same shape of soles and the elasticity of materials such as the polyurethane and others. Finally, finishing the pair of shoes by the surface treatment is performed, placing the insole and arranging the pair of shoes in the box.

The application of inspection systems based on the manufacturing process described above is desirable mainly in two nodes. The first node is after placing the upper on the last, which is performed by a person. This way it would be possible to evaluate the placement of the upper on the last and to inspect if the proper upper is placed on the last and if it is done properly. The most important part of the shoe from the consumer view is the vamp part of shoes because this part of the shoe is the most noticeable, necessitating mirror-symmetrical shoes in pairs and the visibility of possible defects. For these reasons it is necessary to inspect the vamp part of the upper and the placement of the upper on the shoe last. The second possible node would be in case of automation of surplus materials removing from the sole. In this case, the inspection station would perform measurement of the sole surface and evaluation of the correct track of cutting.

The focus is on validating appropriate sensors in inspection systems integrated into systems creating IoT, mainly to compare if it is better to use camera vision or laser sensor in inspection system for shoe production focused on the shoe surface geometry. The next aim is software validation for evaluating the data collected from sensors implemented in the inspection system. For now, camera vision is mainly used in the field of quality inspection and sensors are being integrated into production line monitoring processes.

Based on the description of issues and processes in the shoe production described above, the following two hypotheses were defined:

**Hypothesis** **1** **(H1).**
*Is it feasible to deploy an inspection system evaluating the upper position on the last in the production process?*


**Hypothesis** **2** **(H2).**
*If the deployment of an inspection system is feasible, which method (laser sensors/camera vision) is the most appropriate for the current and future development of inspection and monitoring systems in the shoe production factory?*


## 2. Related Works

Many works dealing with scanning are connected to foot scanning for orthopedic purposes aiming to personalize the development of shoes tailored for a specific person’s foot. Because each person is individual, we have an individual shape of the foot too. For this reason, it was necessary to develop an appropriate device utilizing a specific method. For scanning, it is most necessary to capture a human’s foot and to design the shoe based on this scan [13,14]. The method employs capturing the images by camera vision and laser beam light. The point cloud is computed according to the captured shape of the laser beam by a triangulation method. This method is accurate for generating a point cloud. The same triangulation principle is implemented in laser sensors [15]. The before mentioned method uses the visible spectrum of light. The method of invisible light is based on infra-red (IR) light described in [16] using integrated IR cameras supported by an IR laser sensor in one device.

Usage of the above-mentioned laser sensors is described in [17]. There is a model of the last created by its parametrization. The parameterization is performed over the real last and the real customer’s foot. Measurement is performed by laser scanning. From the scanned data, there is a point cloud generated, which is then transformed into triangle mesh and exported in an *.stl format suitable for 3D printing.

## 3. Materials and Methods

One of the possibilities to use the inspection system described here is monitoring the shoe’s upper position over the shoe last. It is in order to simplify the development of inspection systems, mainly according to expectations from this system. In this case, it is necessary to define the information if there is a proper upper placed over the last and if it is placed correctly. The suitable way of inspecting the placement of the upper is mainly by rotation around the last illustrated in Figure 2 as Y_R_ and displacement in the plane defined by axis X_D_ and Y_D_. From this view of the issue, Y_R_ is the most important value due to the necessity of mirror-symmetrical shoes in pairs and this way evaluate Y_R_ value. Capturing data is suitable in two basic ways and those are camera vision and laser sensors.

### 3.1. The Camera Vision

Camera vision may be applied as a static or a dynamic system capturing the image of the shoe upper placed over the last. The static solution means that a camera is fixed on the pallet with the last and the upper. For this solution, it is necessary to install four cameras, placed on the sides as illustrated in Figure 3. The disadvantage of this solution is the large space demand within the production line. Implementation of this type of stand in an existing line would be complicated due to the need for sufficient space and providing appropriate light conditions including the black box. In the dynamic way, where the shoe last would be placed over a rotary platform, only one camera is sufficient. In this case, the issue with the need for additional free space is solved, but this type of solution is more demanding from a technological point of view because of the need for a more complicated conveyor system or the need to employ a manipulation system for displacing the item to be inspected from the conveyor to the inspection system and back. The other issue is the design of a light system. The shoe industry includes a wide range of types of products from sneakers to boots and a wide range of materials from textile to various types of leathers being used. Setting up an appropriate light condition is very difficult; in this specific case the solution comes up to be very expensive. The other requirement is the necessity of a black box with a door system to prevent light from entering the object’s sensing space by the camera system.

In the field of software, there are few ways of solution possible. The basic solution lays in defining border values. This solution is not adaptable and there is a very high possibility for a wrong decision. Results from this system are very sensitive to input data. Let’s demonstrate it on the input picture shown in Figure 4a. The image was captured with a 12 Mpx camera from Ballufff company (BVS CA-M4112Z00-35-000) with a camera lens, which has a 50 mm focal length (BAM LS-VS-007-C1/1-5018-C). There was a linear monochromatic red light of wavelength 617 nm (BAE LX-VS-LR200-S26) implemented from a distance of 200 mm. It was placed on the left side of the shoe. This implementation produced shadows occurring mainly on the border of the shoe upper. Figure 4b was performed as inverted (*A_opp_*) to the original image (*A_org_*) according to (1) as coded picture by 8 bits, where 0 represents black color and 255 represents white color. This way possibly created an inverted picture, where the black color changes to white and vice versa. The main reason for this procedure was to highlight borders, which were displayed as dark pixels. For this reason, the image shown in Figure 4b is used in further work described in this paper. Figure 4c illustrates dark places from the original image by the usage of the threshold algorithm defined in (2) from Figure 4b. The result is an expression of the most lighted places in the picture, which means edges created by the lighting condition during capturing the image and geometry of the captured object.
(1)$$A_{opp}=255-A_{org}$$
(2)$$B_{i,j}=\left\{\begin{array}{c} 0\;\;\;A_{i,j}<230\\ 255\;\;\;\;A_{i,j}\geq 230 \end{array}\right.$$

It is solvable by edge detection, where the detected edge of the shoe defines its position over the last. Figure 5b illustrates the result of a filter application on the original image shown in Figure 5a. The mentioned filter in Figure 5 was implemented as a matrix of size 5 × 5. Filtering was performed by OpenCV library, working on the convolution principle [18]. This type of filter suppresses the horizontal edge defining the border between the upper and the last. More types of matrices different in both type and size were tested. Results were in general very similar, not enabling us to decide positively about the application of this method. Dispersion in the edges is very wide and it is not possible to define important edges clearly.

A perspective method of edge detection is to use the Canny algorithm [19,20]. This method includes more mathematical operations. For simplification, the Canny algorithm from OpenCV in python language [21] displayed in Figure 6 was used. If comparing Figure 6 to Figure 5, the Canny algorithm suppresses the main features occurring on the shoe. On the other hand, this algorithm much better highlights edges than simple edge detection by the matrix.

The other solution may be considering the implementation of a clustering algorithm on the edges detected in the captured picture. Due to the shape of the edges, the appropriate method is DBSCAN clustering [22,23,24]. To use this method, it was necessary to generate an image with wider detected edges than in previous examples. It was performed by the filter described in (3), that represents the same values in every row, where the first row contains values-15 and the last row contains values 14. The mentioned filter highlights the edges of horizontal character and suppresses other types of edges. The disadvantage is the edges being thick, making this method less accurate. Based on these data it is possible to define the positions of the edges as illustrated in Figure 7. There are highlights in the nine biggest recognized clusters. They are visualized in color, including annotations according to a specific number cluster. This clustering is necessary for the specification of highlighted pixels to the specific detected edge and for identification of required edge characterized border between the shoe upper and the shoe last. Other possible software methods include segmentation etc. Any suitable method from a wide range of available methods and approaches described in scientific papers and in the industry may be used. The method described in this paper using camera vision is suitable for inspection of the position of the upper over the last. Disadvantages are in complication by using blurs, edge detection, and clustering method. Other main disadvantages are inaccuracy and sensitivity of capturing the images by camera vision. The approximate resolution of images captured by camera vision is expressed in (4). Basically, it is the resolution defined in captured images as the specific length visualized (*D*) by correspondent numbers of pixels ($n_{px}$) and decrease resolution by the number of pixels ($A_{i}$) from the matrix used for edge detection.
(3)$$A=\left(\begin{array}{ccc} A_{1,1} & \cdots & A_{1,30}\\ \vdots & \ddots & \vdots \\ A_{30,1} & \cdots & A_{30,30} \end{array}\right)\;where\;A_{i,j}=-\frac{32}{2}+i$$
(4)$$R_{i}=\frac{D}{n_{px}}*A_{i}$$

### 3.2. The Laser Sensors

The method described above based on camera vision is feasible but contains a lot of disadvantages such as low reliability, uncertain evaluation of results, and large space requirements including the black box with door system. The other technology applicable to the inspection issue being described is laser technology. The simplest way is by implementing laser scanners working on the laser triangulation principle [25]. The background of the line laser sensors in the system, where the scanned object is scanned by the camera vision supported by line laser beam and the z-value is computed by triangulation. In [26] Catalog scanCONTROL (2D/3D Laser profile sensors-chapter: The principle of laser line triangulation), where the measuring principle is described based on triangulation, where the emitted line laser beam is reflected onto the sensor matrix as the camera image to calculate the captured profile in the x and z-axis. The third dimension is managed by captured profile-scans in time. The biggest advantages of this method are the much higher resistivity to light conditions during scanning of the object, exact geometric characterization of objects surface, and lower space requirements. The resistivity to light conditions is due to the use of intensive laser beams. The exact geometric characterization is due to scanning a surface and interpreting it as the point cloud. Capturing as the point cloud enables mathematic methods of source data evaluation to be used and exact results to be generated by usage in the manufacturing process. The disadvantage, when compared to the camera vision, is in colorless captured data, only available in grayscale. The lower space requirements come from the working principle of the laser sensor and the compact size of the laser sensor body. For laser sensor implementation, sensors from Micro-Epsilon company were chosen, because of previous authors’ work [27]. There are also other manufacturers of laser sensors on the market.

In Table 1, there are various types of laser sensors produced by Micro-Epsilon company specified. This manufacturer offers more types of laser sensors including standard red laser and special blue laser sensor, types 29xx and 30xx. Type highspeed sensors are named xx50-xx, which are adapted to the possibility of high-frequency scanning. Other types are SMART sensors 251x, 261x,291x, and 301x. Other types available are high-speed smart sensors 266x, 296x, 306x [15].

Explanation of values after the dash in type name of the sensor is defined in Table 2, where measuring ranges in the Z-axis are defined [28,29,30]. Mentioned information in Table 1 and Table 2 are important from the view of scanned objects. The scanned objects or the shoes can be simplified to declare basic scan size requirements. The majority of shoe production varies in size from 33 to 50 in EU size standards. It represents the dimensions of shoes in length from 200 to 330 mm and width from 50 to 120 mm. The other dimension is height, which is mainly from 50 to 80 mm. In case of the maximal reachable obtained resolution by the laser sensors, it is possibly defined based on avalibities of used hardware devices.

After specifying the available laser sensors from the Micro-Epsilon company and defining the basic dimensions of the majority of shoes being produced, it is necessary to analyze the scanning method. There are three surfaces to be scanned illustrated in Figure 8. The first surface represents the place of a heel part and the lower part of the shoe last. For scanning in a sufficiently low measuring range, 50 mm is suitable according to Table 2. 

These sensors have a higher resolution which enables scanning with higher precision. Another advantage is in scanning the last, enabling modification of the results and consideration of the position of the last and based on this information modification of the mathematical model for evaluating the position of the heel part. The second way is scanning the place of the vamp part and the front part of the last. It is appropriate according to being the most conspicuous part of the shoe. From the aesthetic point of view, it is a very important part of the shoe as a product of the fashion industry. But for the scanning it is necessary to use laser scanners with a larger measuring range. The last one of possible scanning approaches is scanning from the bottom of the shoe upper. This surface is characterized by flat topology. For such a scan a laser sensor with a wider measuring range in the x-axis [30] is appropriate. That is why it is necessary to use a sensor with a measuring range of 50 or 100 mm. The character of the surface is simple to scan and evaluate. It does not have a large impact on the final product. To sum up, in further work it is appropriate to scan the place of the vamp part of the shoe, as the important part of it.

In Figure 9, the concept of the inspection system is displayed. It was designed for the possibility to scan a heel part and a vamp part of the shoe upper. Specifically, it was designed for the static measuring test, where two types of laser scanners were used, as defined in a figure per each scanning surface. The concept is demonstrative for the size of the shoe 37. Linear axes with laser sensors are adapted to capture larger space including the size of shoe 50. Red rectangles represent basic measuring fields, which are achieved by linear axis and laser sensors configuration.

### 3.3. The Experimental Inspection Stand Based on the Laser Sensor

In Figure 10, an experimental inspection stand is shown. When compared to the concept, the experimental stand lacks one laser sensor, which is due to its financial expensiveness. Before developing the final device, it is necessary to perform a validation test. For this purpose, this experimental stand is sufficient both from the hardware and software view. In case of hardware, necessary conditions for experimental stand validation are dimensions of the construction, measurable space, and modularity. Modularity includes the ability of reconstruction and adaptation when implemented in the production line and the possibility of increasing the number of sensors for measuring both pieces of a pair of shoes at the same time. For construction, the usage of struts profiles is available. On this stand, the static test of software design was performed. Validation of the sensor includes the possibility to capture an inevitable surface, enough points in a line. For sensors 2950-100, the maximum measurable number of points in one line is 1280 which represents resolution in the X-axis according to scanned length. The basic plane is XY consisting of the X-axis representing the position of points in line and the Y-axis represents such captured lines in time. Resolution in the Y-axis is represented in the number of line scans per second and tracks performed by the linear axis. Resolution on the X-axis ($R_{x}$) is definable such as length of polyline from point to point ($X_{a}$ and $X_{b}$) expressed in (5), where the *f*(*x*) is the function obtained by polynomial regression of points in a specific line. $X_{a}$ and $X_{b}$ represent position of the first and the last point of function for calculation of resolution on specific place “b” and “a”, which represents position of points captured line by a laser sensor. $P_{i}$ represent the position of points in line expressed in (6). The algorithm based on (6) is used for sorting points in the matrix according to their X-value. $X_{gmin}$ and $X_{gmax}$ represent global extreme values captured in a specific scan.
(5)$$R_{x}=\frac{\int_{X_{a}}^{X_{b}}\sqrt{1+{\left(f^{\prime }\left(x\right)\right)}^{2}}dx}{b-a};\;a,b\;\epsilon \;P_{i},\;\;$$
(6)$$P_{i}=\left\lfloor \frac{X_{i}-X_{gmin}}{X_{gmax}-X_{gmin}}\right\rfloor ;\;P_{i}\epsilon Z;\;i\epsilon \langle 1,1280\rangle$$

Resolution on the Y-axis ($R_{y}$) is definable as the length of polyline from point to point ($Y_{a}$ and $Y_{a}$) according to the time of scanned profiles expressed in (7). The $f\left(y\right)$ is function obtained by polynomial regression of points defined in specific $P_{i}$-position (X-axis) with coordinates Z a Y. The Y coordinates corresponding to the time of captured points. The specific move length is defined as the function of moving the laser sensors, object, or both during the scanning divided by nominal frequency. During the experiments the real frequency floating around the nominal frequency mainly in ±5% of the nominal frequency.
(7)$$\;R_{y}=\frac{\int_{Y_{a}}^{Y_{b}}\sqrt{1+{\left(f^{\prime }\left(y\right)\right)}^{2}}dy}{\frac{\int_{t_{a}}^{t_{b}}f\left(t\right)\;dt\;}{f}}$$

Theoretically the best achievable resolutions of flat surface for 2950-100 sensors in scanning distance 226.4 mm in the Z-axis are definable based on the datasheet. The resolution ability in the Z axis-$R_{z}$ is defined as 0.012 mm. Line linearity is defined as ±0.012%. The maximal deviation of a single point is ±0.10% [29]. In [29] scanning trapezoid for computing the length of the line in a specific distance in the Z-axis is defined. For distance 226.4 mm in the Z-axis is the length of line 94.17 mm. The resolution in the X-axis $R_{x}$ for 1280 points is 0.0735 mm. The resolution for the Y-axis is defined based on the velocity of linear motion as 20 mm per second and nominal scanning frequency 100 Hz is $R_{y}$ = 0.2 mm.

Software for evaluation of the shoe upper position over the shoe last is possible to be reached in more ways. The basic way is a comparative method based on the comparison of the etalon model and scanned model to be inspected. This method is simple and does not require high computing power. Its disadvantage is the absence of analytical output information. The system can declare if it is positioned properly or not. In case of requirement of the information about displacement or wrong parameters of the shoe upper placement, some other method of evaluation needs to be used. In the comparative method, the etalon model can be created as a median scan from more correct scans.

In Figure 11, there are scans of correct placement of the upper over the last displayed. These data are the reference for the generation of etalon data (Figure 11b). Data are in form of a point cloud [31,32]. For the generation of etalon data, it is necessary to sort the data to evaluate positions and to define the appropriate position of points in the space. This is done mainly by generating the algorithm for sorting points based on (6), summarizing them, and computing the median in the specific position in (8), where “k” represents the number of scans used to generate the etalon data. Variable “n” represents the numbers of $A_{i,j}$ from scans for computing the mean value of etalon data $\left(M_{i,j}\right)$.
(8)$$\;M_{i,j}=\frac{\sum_{1}^{k}A_{i,j,k}}{n}\;if\;A_{i,j,k}>0;\;$$

The comparative method works on computing the distance of the scanned point cloud from etalon data. The main parameter is the number of points with a larger distance then is the allowed value or the threshold. These parameters are defined on specific requirements of manufacturers according to quality parameters. Output from this method is the information if the upper is placed correctly or there must be correction performed. In the case of if correction is necessary, it is performed in a way of replacing the upper on the last because there is no information about the correction to be performed. So, this method is poor in data mining for factory implementation for instance IoT or Industry 4.0.

In Figure 12, there is a comparison of the test scan with the etalon scan displayed, where the etalon scan is shown in green color. The test scan being evaluated contains black points representing not-captured points, red points represent points with a larger distance between inspected scan and etalon test than is the threshold value. Points displayed in grey are in tolerance position defined by the threshold. According to the number of red points, a statement can be made about the placement correctness of the upper or necessity to perform correction by replacing the upper over the last. In case of expression differences from the etalon scan, it is necessary to design algorithms for computing the parameters of displacement.

As shown in Figure 2, the main parameter is YR, describing rotation and difference can be computed as an angle between the tangents. Expressing tangents is suitable through polynomial and tangent to a specific point. Polynomial is computable by numpy library in python [33]. There is the possibility to compute a polynomial of one to seven degrees. For uneven surfaces, it is better to use a polynomial of higher degree for better approximation to the real position of points in the line. Polynomial regression is performable in two ways in line, where the first case is a regression of the Z-values and positions of correspondence points in the line. However, it is suppressing positions of points in line defined by X-coordination. The advantage is in simpler computing. The second is polynomial regression based on Z and X coordination of points in line as is defined in (9). This method is more precise to represent point cloud as surface captured by the laser sensor.
(9)$$\beta_{y}={\left(X^{T}X\right)}^{-1}X^{T}{\bar{z}}_{my}$$
(10)$${\bar{z}}_{my}=({\bar{z}}_{y}>0)$$

For regression, it is necessary to modify the input data because input vectors ${\bar{z}}_{y}$ contain zero values for non-captured points. While computing regression including these zero values, polynomial would be jumping to zero values in line. This regression was being useless for further works. For this reason, an algorithm is implemented to separate points with Z-values higher than 0 mentioned in (10). After obtaining regression polynomial representing points in line, the first derivation of the polynomial is generated. Based on derivation (11) it is possible to calculate the equation of the tangent line (12) to the mentioned regression polynomial. There is needed a tangent point necessary to be defined and obtain its coordinates ($x_{i},z_{i}$), as $z_{i}$ defined in (13), basically $z_{i}$ is the highest Z-value in the specific line of the scan. The last step is to express an angle between tangents of etalon scan and inspected scan in (14), where $\varphi_{e}$, $\varphi_{t}$ are angles of tangents of etalon scan and inspected scan with basic plane XY.
(11)$$Q=\frac{\partial \beta }{\partial x}$$
(12)p=kx+q
(13)$$k=Q\left(x_{max}\right)=Q\left(x_{i},z_{i}\equiv \max\left(Z\right)\right)$$
(14)$$\varphi =\varphi_{e}-\varphi_{t}=\tan^{-1}\left(k_{e}\right)-\tan^{-1}\left(k_{t}\right)$$

That way it is possible to evaluate the position of the upper and to express the wrong position of the upper. In Figure 13, an inspected scan (a) is displayed with tangents by red color. On the right side (b), there is the generated etalon scan with tangents displayed.

Summarizing the above-performed work is in Figure 14, where it is displayed combining the etalon scan with tangents in green color and inspected scan by visual red and black color and correspondent tangents. The left side of Figure 14 demonstrates the rotation-Y_R_ in a specific way, where one part is significantly rotated in one direction. The right side of Figure 14 demonstrates the dominating of the green color of etalon scan and inspected scan in red color is under the green surface.

The number evaluation of angles is illustrated in Figure 15a. There is a significant number of angles from zero to four degrees of rotation in Y_R._ In Figure 15b there are approximated points to a polynomial of 7th degree with computed standard deviation 2.3019°.

## 4. Results

In This paper, there are two methodologies of performing inspection issues for the shoe industry described. In the first case, it was the method of capturing 12 Mpx images by camera vision. These data were used to evaluate the position of the upper on the last. Evaluating was performed by processing image, edge detection, and final DBSCAN clustering, separating white pixels to edges. To sum up, there is the possibility to inspect an object’s position as geometric characterization by camera vision. In case of the requirement of higher precision than 1 mm, it is complicated due to basic resolution characterized as the captured length in the corresponding number of captured pixels in (4). Resolution this way of the mentioned camera device is approximately 0.1 mm. In combination with a large size matrix filter, defined in (3), resolution decreases to approximately 3 mm with high uncertainty. Using camera vision in the shoe inspection system is appropriate in visual surface inspection [34]. On a geometric surface, shoe inspection is feasible but unreliable. The second mentioned method using laser sensors is more satisfying in resolution, device requirements, and reliability. Resolutions are described by (5) for X-axis and (7) for Y-axis, which are according to the scanned shape surface approximately from 0.2 to 0.5 mm per point (mm/px). This correspondence to real obtained resolution according to the type of sensing device-sensors and scanned surface and their common positions. The best reachable resolutions in ideal conditions were expressed as $R_{z}$ = 0.012 mm in scanning distance 226.4 mm, $R_{x}$ = 0.0735 mm and $R_{y}$ = 0.2 mm and final resolution R = 0.2134 mm. Evaluation of the resolution based on (5) is displayed in Figure 16. The resolution in the Y-axis (7) is displayed in Figure 17. Evaluation is shown on Scan 20, which represents file 2020_09_21__00_0__2__0__1__11.csv. For this shoe type we performed 67 experimental scans which have been evaluated to simulate any wrong placement positions. For the resolution in the Y-axis the parameters of scanning are important, where the scan frequency was set to 100 Hz and line velocity of linear motion was 20 mm/s. The main core is computed as polynomials of 6th degree by polynomial regression as $f\left(x\right)$ from a specific part of point cloud defined by shoe scan. Based on the resolution on the X-axis and on the Y-axis the resolution of point cloud can be computed, which is displayed in Figure 18, where the results are expressed as the square root of the sum of the squares of the individual resolutions. Evaluation resolution values are displayed in Figure 19. There are also higher values occurring than is displayed in mentioned images. These values are mainly in point poor areas in the point cloud and in hard-scannable areas, which generate quite different values from the main values in the range from 0.001 to 0.5 mm per point. The higher values were threshold to the maximal displayed value.

According to hypothesis 1: Confirmed, it is possible to applicate the shoe inspection system to evaluate the shoe upper position over the shoe last in the production process. Hypothesis 2: The more appropriate is the shoe inspection system based on the laser sensor, where it is possible to achieve the real resolution of 0.5 mm per point. In the symmetry faces of laser sensors, the resolution from 0.16 to 0.3 mm per point is achieved. For inspection of the geometric placement of an object over the last only, it is more suitable to build a shoe inspection system employing a laser sensor. In the case of a combination of visual and geometrical inspection tasks, an answer is more complicated, and the solution depends on complex requirements from manufacturers, that want to integrate the inspection system. For automation of the repositioning of the upper over the last, it is necessary to apply an analytical method of shoe inspection system described in Section 3.3. This method is also built on polynomial regression defined by polynomials and based on a specific point where it is possible to define vectors of tangents. According to these tangents, it is possible to compute the angles between tangents of the inspected scan and etalon scan. For instance, evaluation of these tangents for Scan 20 is displayed in Figure 14 and Figure 15.

## 5. Discussion

In this paper, there are two methods for geometrical shoe inspection systems described. The measurement by laser sensor and image capturing by camera vision was performed. Results indicate a significantly more satisfying output in the case of laser sensor than in the case of camera vision. To inspect the geometrical character of the surface it is necessary to capture this information by laser sensor for evaluating the shoe upper placement as part of quality inspection in production. This shoe production is very sensitive to high quality due to the fashion field. For this reason, the manufacturers are focusing on the development of Industry 4.0 and IoT methods imposing requirements to collect data from production systems, material flow, output quality inspection, etc. According to this, the data collection is performed by complex sensor systems implemented in every possible usable place for the improvement of production parameters. In this way, it is an effort to achieve a better position in the global market and to be more competitive.

From the view of a designed shoe inspection system, there is potential for the development of systems based on the laser sensors implemented in the production system. The obtained data from these sensors are in the form of a point cloud, which contains very precise data from the geometrical view of the inspection issue. The evaluation of point cloud was performed by the polynomial regression in two areas. The first was polynomial regression to evaluate resolution possibilities according to scanned surface. Thanks to the described method, it is possible to focus on further work on the possibility of the inspection system for defect detection and pattern recognition, where there is an emphasis on high resolution of the scanned object. Thanks to described methods it is possible to continue development in other areas and to decide if is the geometrical data are of necessary quality and resolution. In the case of the shoe scanning, the resolution from 0.16 to 0.5 mm per point was obtained. In the vamp area of the upper, this resolution was in the range of 0.16 to 0.25 mm per point. In this way, it is possible to think about developing an inspection system for defect detection based on geometrical data. The second area implementing the polynomial regression was in the evaluation of Y_R_ by angles between the tangents as is displayed and described in Figure 14 and Figure 15. Based on this information the automation system with a procedure to revise misplacement of the shoe upper on the shoe last can be built.

In the introduction it is mentioned the commercial solution of scanning devices from companies Zivid and Photoneo. In this work is not a comparison between the application of 3D scanners such as Zivid Two or PhoXi 3D Scanner GEN 2 working on the camera vision and 3D line scanners from Micro-Epsilon company used in this paper. In the case of Zivid One+ Small it is declared that the point precision in Euclidian space is 25 μm (Datasheet [35]) in work distance approximately 0.370 m. We declared the spatial resolution in approximately distance 0.370 m is 0.13 mm (values obtained from graphs in the datasheet). In the case of the Photoneo company, the manufacturer declares for PhoXi 3D Scanner S point to point distance 0.17 mm in sweet pot 442 mm (datasheet PhoXi 3D Scanner S Generation 2 [36]). Furthermore, the Micro-Epsilon company offers surfaceCONTROL 3D 3500, where for type SC3200-80 is declared resolution for X and Y-axis from 0.055 to 0.070 mm and in Z-axis 0.0015 mm in the working distance in the Z-axis 130 ±10 mm (Datasheet surfaceCONTROL 3D 35xx/32xx [37]). The above-mentioned solutions are possible to use to shoe position inspection system. Further work should perform experiments with this system and compare it to the method demonstrated in this paper based on 3D line laser sensors. Declared values are in good condition. This paper was focused on the resolution on the general surface, where it was confirmed a high dependency of shape on the scanned surface to achieve resolution. Resolution is also possible to separate into three categories for each axis X, Y, and Z. The line laser sensors achieve great resolution in the Z-axis. The resolution in X and Y-axis is comparable to the above-mentioned 3D camera vision. But the big advantage of 3D camera vision is that these systems do not require specific movements of sensors such as the line laser sensors and color surface scanning.

In the case of the scanning of variable types of materials, in this paper, the experiments were performed on the textile upper. The next work should be appropriate to confirm the possibility of scanning transparent material. Based on documentation from the manufacturer, for these purposes the laser sensors with blue light beams are recommended. These types of laser sensors are recommended for scanning the material, where light penetrates to this material or for organic material. The other application is for red-hot material. These possibilities are based on the use of blue light with a wavelength of 405 nm, which has bigger energy and a shorter wavelength compared to the red light with a wavelength of 658 nm. The impact of vibration from the factory environment was not explored due to being performed only in laboratory experiments, which were performed out of the manufacturing factory in static mode. In the process of design, inspection stands have been minding the impact of vibration. For this purpose, the inspection stand was adapted to scan the upper and the shoe last. Based on data from the scan of the shoe last, suppose the design automated system of cleaning data for everyone scan separately. The inspection stand was designed to possibility assemble the linear axis with laser sensors for scanning of the pair of the upper on the shoe last in one cycle. In this way, there occur the interference in scanning due to reflected light beam from different laser sensors. The laser sensors contain RS422 port, which is usable for synchronization several sensors.

## Figures and Tables

**Figure 1 sensors-21-07531-f001:**
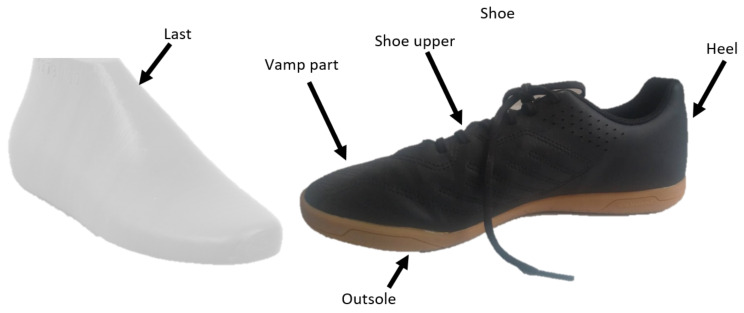
The basic parts of the shoe.

**Figure 2 sensors-21-07531-f002:**
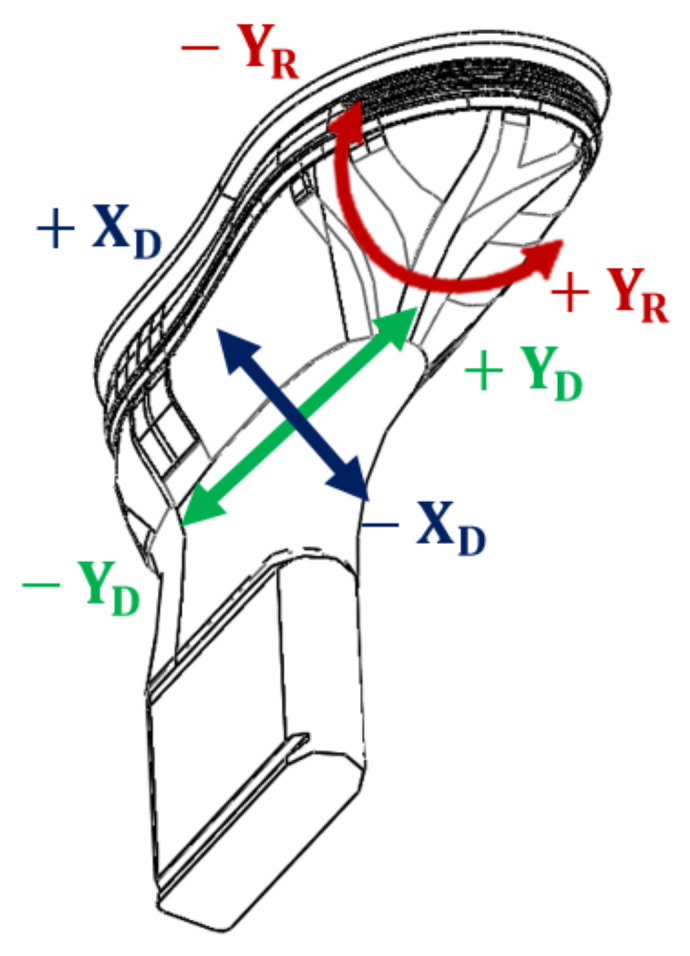
The possible wrong placements.

**Figure 3 sensors-21-07531-f003:**
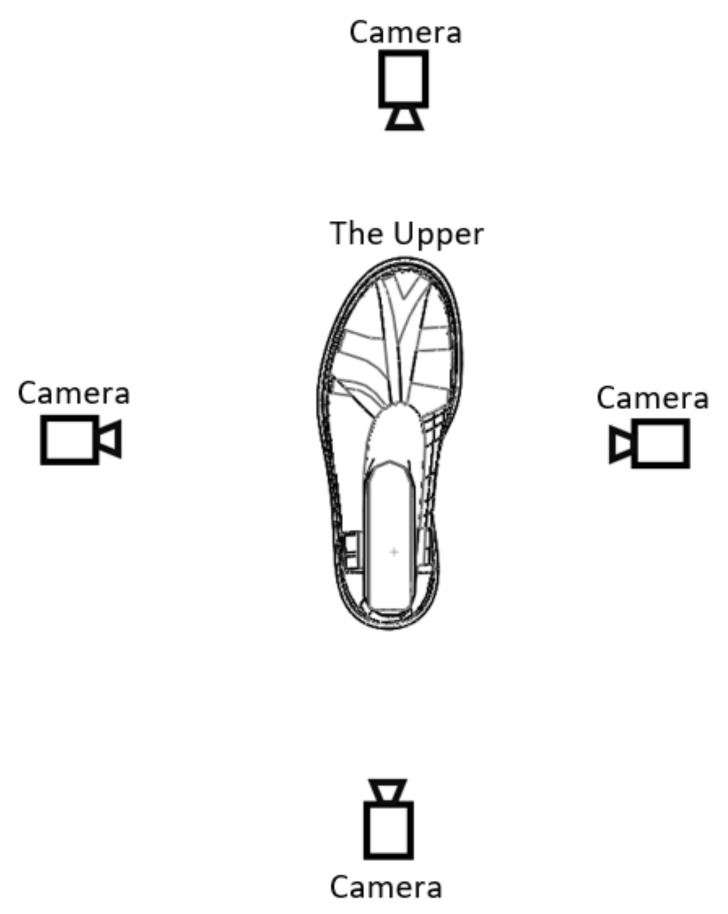
Placement of cameras.

**Figure 4 sensors-21-07531-f004:**
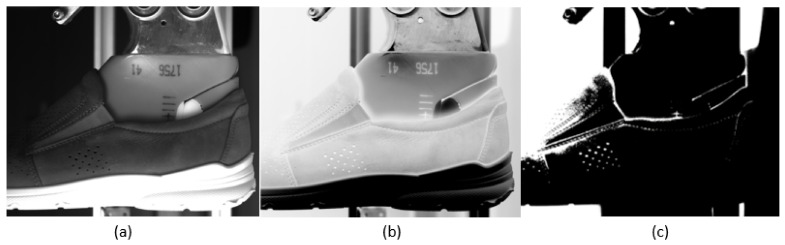
Illustration of original picture (**a**), inverted picture (**b**) and highlight shadows (**c**).

**Figure 5 sensors-21-07531-f005:**
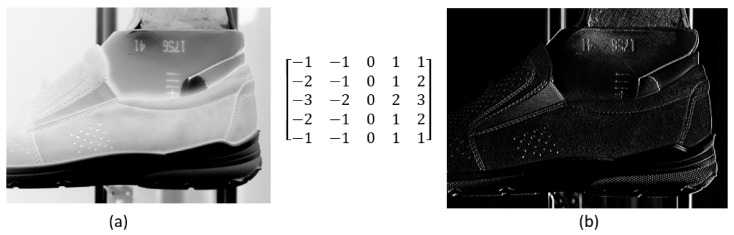
Original (**a**) and filtered (**b**) picture.

**Figure 6 sensors-21-07531-f006:**
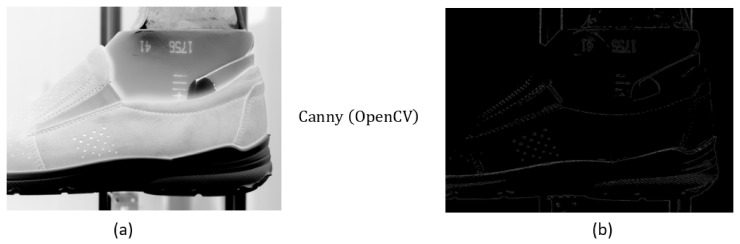
Original (**a**) and filtered by Canny algorithm (**b**) picture.

**Figure 7 sensors-21-07531-f007:**
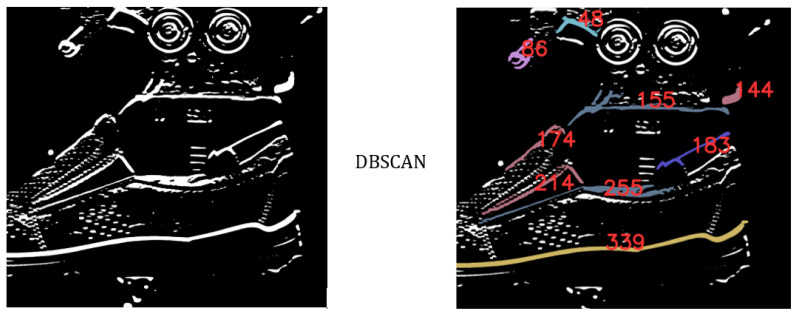
Implementation of the DBSCAN clustering to edges.

**Figure 8 sensors-21-07531-f008:**
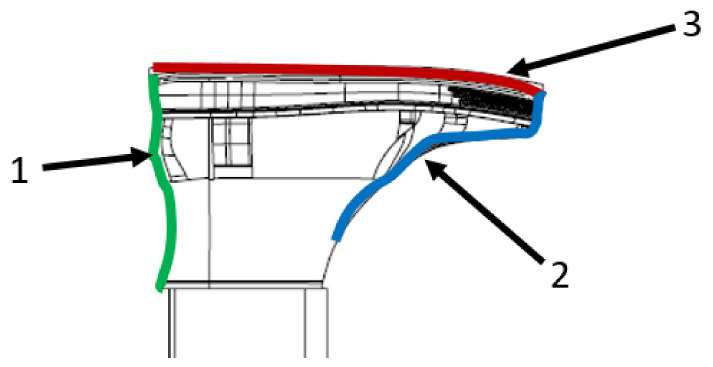
Scannable surfaces on the shoe.

**Figure 9 sensors-21-07531-f009:**
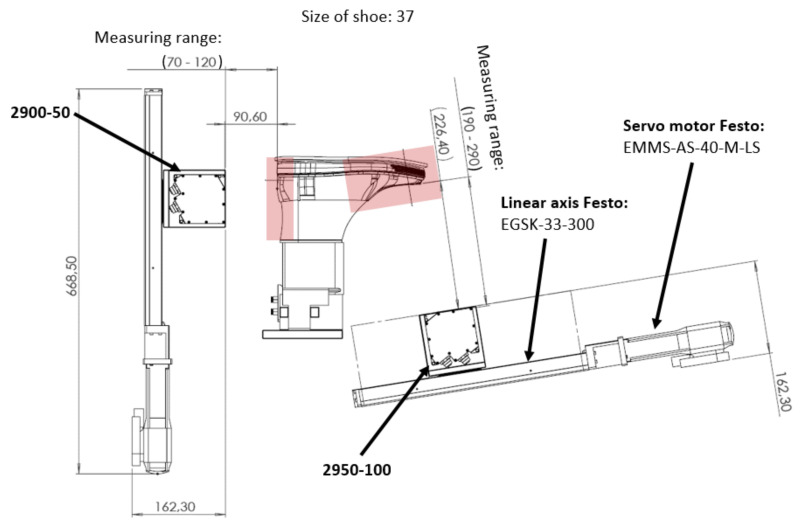
Concept of inspection stand.

**Figure 10 sensors-21-07531-f010:**
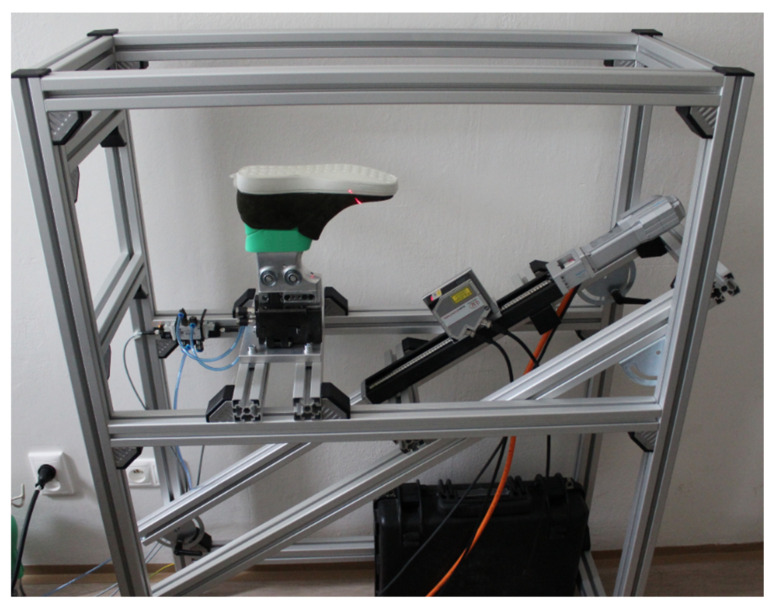
Experimental stand in laboratory.

**Figure 11 sensors-21-07531-f011:**
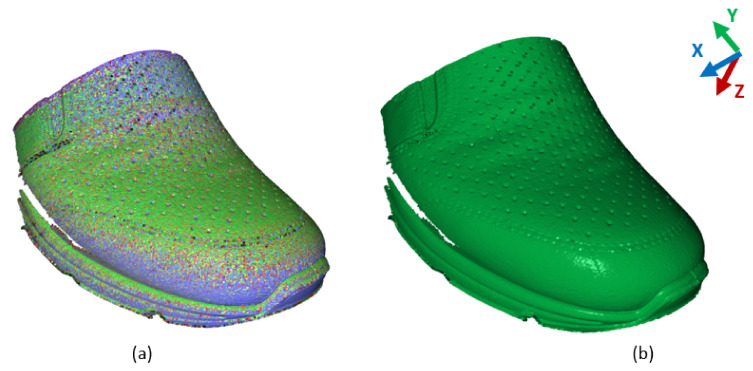
Scans of correct placed (**a**) and generated etalon scan (**b**).

**Figure 12 sensors-21-07531-f012:**
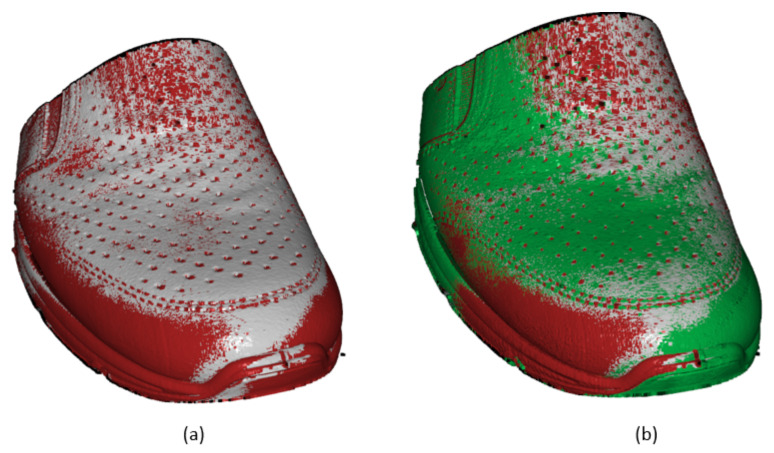
Inspected scan 10 (**a**) and inspected scan with etalon scan (**b**).

**Figure 13 sensors-21-07531-f013:**
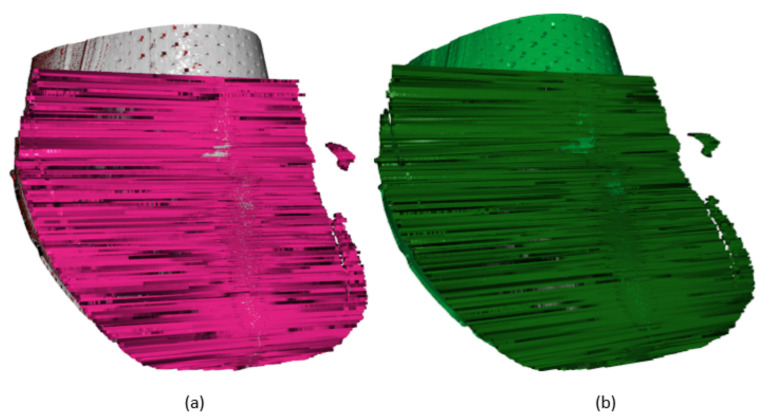
Inspected scan with tangents (**a**) and etalon scan with tangents (**b**).

**Figure 14 sensors-21-07531-f014:**
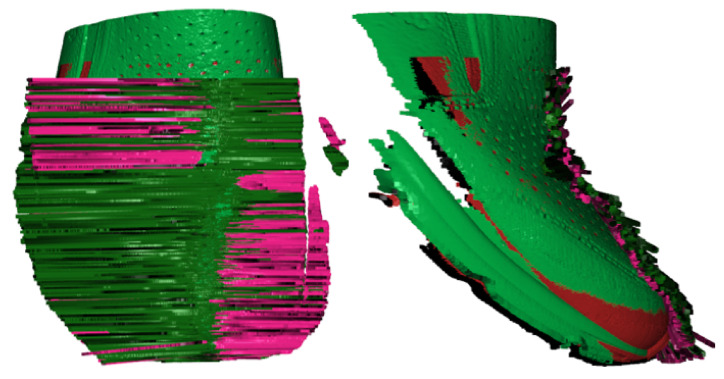
Comparison etalon scan with inspected scan.

**Figure 15 sensors-21-07531-f015:**
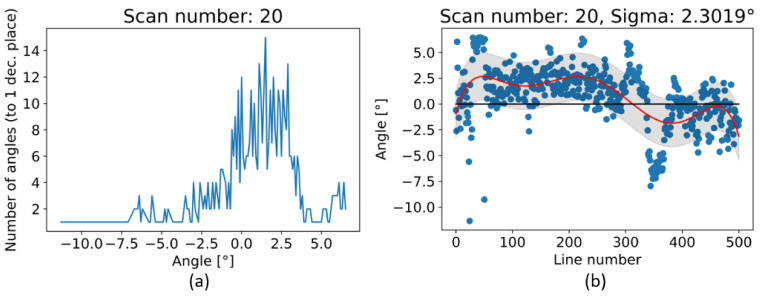
Angles between the tangents for scan number: 20, (**a**) number of angles rounded to one decimal place and (**b**) angles approximated to polynomial of 7th degree with standard deviation.

**Figure 16 sensors-21-07531-f016:**
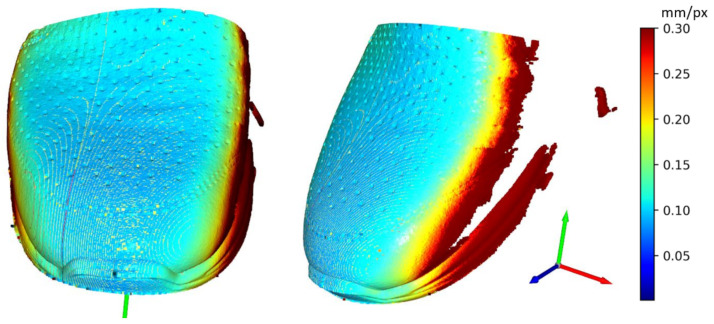
Displaying of resolution (Rx) in X-axis (SCAN 20).

**Figure 17 sensors-21-07531-f017:**
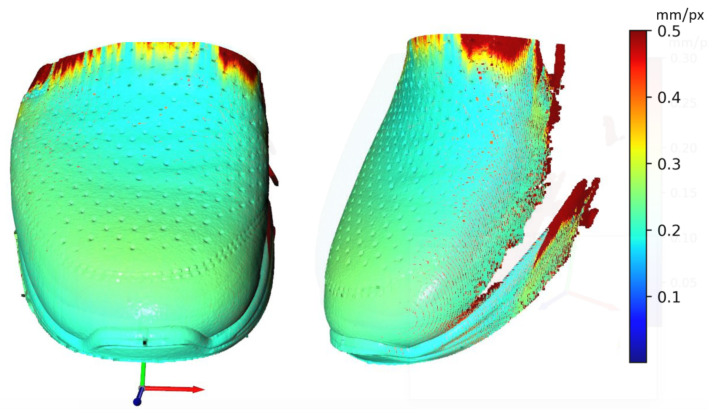
Displaying of resolution (Ry) in Y-axis (SCAN 20).

**Figure 18 sensors-21-07531-f018:**
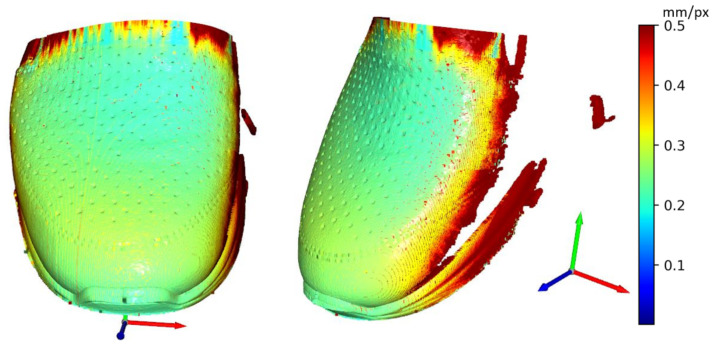
Displaying of resolution (R) (SCAN 20).

**Figure 19 sensors-21-07531-f019:**
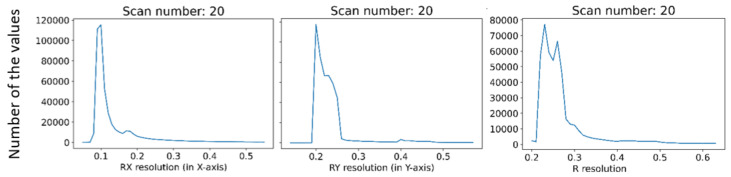
Resolutions for Scan 20.

**Table 1 sensors-21-07531-t001:** Laser sensors Micro-Epsilon.

Laser Sensor Type scanCONTROL	Number of Points in One Line (Max)	No of Profiles per Second Up to (Hz)	Resolution in Z-Axis (10^−6^ m) (Type-Model)
25xx	640	300 (standard)	2 (2500-25) 4 (2500-50) 14 (2500-100)
26xx	640	300 (standard)	2 (2600-25) 4 (2600-50) 14 (2600-100)
		4000 (highspeed)	2 (2650-25) 4 (2650-50) 14 (2650-100)
29xx	1280	300 (standard)	2 (2900-25) 4 (2900-50) 14 (2900-100)
		2000 (highspeed)	2 (2950-25) 4 (2950-50) 14 (2950-100)
30xx	2048	300 (standard)	1.5 (3000-25) 3 (3000-50) 26 (3000-200)
		10,000 (highspeed)	1.5 (3050-25) 3 (3050-50) 26 (3050-200)

**Table 2 sensors-21-07531-t002:** Measuring ranges of specific sensor types.

Standard Measuring Range in Z-Axis	25xx, 26xx, 29xx-25	25xx, 26xx, 29xx-50	25xx, 26xx, 29xx-100	30xx-200
Start of measuring range	53.5 mm	70 mm	190 mm	200 mm
Mid of measuring range	66 mm	95 mm	240 mm	310 mm
End of measuring range	78.5 mm	120 mm	290 mm	420 mm
Height of measuring range	25 mm	50 mm	100 mm	220 mm

## Data Availability

The data presented in this study are available on request from the corresponding author.

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
