# Peer review of "Analysis of Laser Sensors and Camera Vision in the Shoe Position Inspection System"

_sensors, 2021, doi:10.3390/s21227531_

Round 1

Reviewer 1 Report

The authors of the article presented a qualitative description of the process of camera vision and laser sensors applied to the shoe industry. Point of clouds is collected with a resolution up to 0.16 mm per point. The results are directly related to the development of industry 4.0.
The manuscript is well structured. Before publishing, several questions should be resolved:

  1. The abstract contains links to formulas that the reader can find in the article itself. This is invalid for annotation. Try to describe it if necessary.
  2. The beginning of the introduction contains facts about the development of Industry 4.0. However, there are no supporting references. The reader is left to take the author's word for it. I recommend adding a few modern references to this topic.
  3. The authors consider machine vision using 4 cameras. Nowadays, there are simpler-to-implement technologies using 1 camera to build a point cloud. A good example is a commercial product from Zivid (https://www.zivid.com/zivid-one-plus). Sometimes they can be found in scientific publications (DOI 10.1109 / ISMR48331.2020.9312948).
  4. In Figure 4, the second photo is labeled "oposite picture". Most likely the authors mean an inverted image?
  5. If camera vision works through algorithms and signal processing, then what physical principles are used in laser sensors? The authors assure that higher sensitivity is achieved. However, the intensity of laser radiation is a very relative concept. Most likely, the point is not in the amplitude of the signal, but in the phase?
  6. Table 1 contains a number of values, but no dimension is indicated for them. It is worth correcting the table. Also not clear "standard" and "highspeed" - is it modes or smth else?
  7. What happens if part of the boot turns out to be made of optically transparent material? Is this a limitation of the applicability of the method?
  8. The experimental setup is assembled from a metal profile. How much will vibration in a room or as a result of linear actuators distort the measurement results?
  9. Figure 15 is an experimental result? How correct is it to connect these points with a solid line?

Author Response

Dear Reviewer,

Thank you very much for your comments. The attached file lists the paragraphs that have been modified or added in the manuscript. We believe that the answers will be relevant for you. Otherwise, please write additional comments or questions.
Thanks again for reviewing the manuscript.
Yours sincerely
Authors

Reviewer 2 Report

The whole research progress is useful for industry. But there isn't enough significance for a scientific paper.

The following is my comments:

  1. In this paper, there is no innovative method or algorithm to employ. The results are not good enough to highlight the scientific significance of this research.  
  2. The parameters and equations are not explained completely and clearly in this paper.
  3. The resolutions are defined by equations (5) and (7), why? It isn’t clear what the meaning of the resolution is.
  4. Which parameters of shoes are needed to be measured?
  5. The measurement precision must be analyzed in this paper.
  6. Which method is used to reduce random inspection interference?
  7. Are there any other inspection technologies used to compare with the methods in this paper?
  8. What’s meaning of “scan 20” in figures?
  9. In this paper there is comparison between camera equipment and laser sensors.  But the tasks of them are different, one is edge detection and the other is to get the resolution. Is there any connection between them?  

Author Response

(The authors gave the same response as above.)

Round 2

Reviewer 2 Report

It is a good work for shoe industry. I hope more research on how to improve  its practicability and efficiency.